# Inter-Individual Variability in Metabolic Syndrome Severity Score and VO_2_max Changes Following Personalized, Community-Based Exercise Programming

**DOI:** 10.3390/ijerph16234855

**Published:** 2019-12-03

**Authors:** Sophie Seward, Joyce Ramos, Claire Drummond, Angela Dalleck, Bryant Byrd, Mackenzie Kehmeier, Lance Dalleck

**Affiliations:** 1Recreation, Exercise & Sport Science, Western Colorado University, Gunnison, CO 81231, USA; sophie.seward@western.edu (S.S.); adalleck@western.edu (A.D.); bbyrd@western.edu (B.B.); mackenzie.kehmeier@western.edu (M.K.); 2SHAPE Research Centre, Exercise Science and Clinical Exercise Physiology, College of Nursing and Health Sciences, Flinders University, Adelaide 5024, Australia; joyce.ramos@flinders.edu.au (J.R.); claire.drummond@flinders.edu.au (C.D.); 3Centre for Research on Exercise, Physical Activity and Health, School of Human Movement and Nutrition Sciences, The University of Queensland, Brisbane 4072, Australia

**Keywords:** cardiovascular disease risk, cardiorespiratory fitness, metabolic syndrome, type 2 diabetes, responders, training responsiveness

## Abstract

This study sought to examine the effectiveness of a personalized, community-based exercise program at reducing MetS severity and consequently Type 2 diabetes mellitus (T2DM) and cardiovascular disease (CVD) risk. One-hundred and fifty physically inactive participants (aged 18–83 years) were randomized to a non-exercise control group (*n* = 75; instructed to continue their usual lifestyle habits) or treatment group (*n* = 75). Participants randomized to the treatment group completed a 12 week personalized exercise training program based on the American Council on Exercise (ACE) Integrated Fitness Training (IFT) model guidelines. Z-scores were derived from levels of metabolic syndrome risk factors to determine the severity of MetS (MetS z-score). After 12 weeks, the treatment group showed a significant favorable change in MetS z-score, whereas the control group demonstrated increased severity of the syndrome (between-group difference, *p* < 0.05). The proportion of MetS z-score responders (Δ > −0.48) was greater following the exercise intervention (71%, 50/70) compared to control (10%, 7/72) (between group difference, *p* < 0.001). The inter-individual variability in VO_2_max change also showed a similar trend. These findings provide critical translational evidence demonstrating that personalized exercise programming based upon the ACE IFT model guidelines can be successfully implemented within the community setting to reduce T2DM and CVD risk.

## 1. Introduction

Type 2 diabetes mellitus (T2DM) significantly increases an individual’s risk of cardiovascular disease (CVD) [1] and is among the top 10 global causes of deaths [2]. The prevention of this condition is therefore considered a public health priority [2]. However, common biomarkers used to track T2DM risk reduction during treatment interventions have been challenged [3]. It has been recently reported that a continuous measure of metabolic syndrome (MetS) severity (MetS z-score) is a useful biomarker to help track earlier responses to treatment, which may provide greater adherence to exercise program interventions specifically targeted towards the prevention of T2DM and CVD [4]. Indeed, exercise-induced cardiorespiratory fitness (CRF) has been demonstrated to be an antidote against the severity of MetS [5]. Barry et al. [5] showed a strong inverse association between MetS z-score and CRF, with those with higher CRF demonstrating lower MetS z-score. MetS z-score appears to be sensitive to exercise interventions [6,7,8,9], with a more ‘personalized’ approach to exercise prescription showing reduced inter-individual ‘MetS severity’ changes, depicted by a high proportion of individuals meeting the threshold of change that is deemed to be a clinically meaningful change beyond one’s biological variability (so called ‘responders’) [8]. 

There is limited available research on the impact of community-wide exercise initiatives on primordial and primary prevention of T2DM and CVD. Nevertheless, our research group has previously demonstrated that a community-based exercise program is an effective model to lower CVD risk in adults by markedly reducing MetS prevalence and abolishing individual MetS components [10]. These prior experimental findings provide preliminary evidence for the implementation of widespread primordial and primary prevention strategies focused on increasing physical activity and exercise in order to mitigate the burden of cardiometabolic disease in the community. However, previous research has been hampered by lack of a personalized approach to the exercise prescription, which has been previously demonstrated to result in non-responsiveness and adverse responses to exercise training [11]. There is a critical need for additional research that can provide the important translational evidence needed to implement personalized, evidence-based exercise programming into the community to reduce MetS severity and prevent T2DM and CVD. The purpose of this study was to examine the effectiveness of a personalized, community-based exercise program at reducing MetS severity.

## 2. Materials and Methods 

Nonsmoking men and women (*N* = 150, 18–83 years of age) were recruited from a local university and surrounding community via advertisement through the university website, local community newspaper, and word-of-mouth. Participants were eligible for inclusion in the study if they were low-to-high risk as defined by the American College of Sports Medicine and not physically active [12]. This study was approved by the Human Research Committee at Western Colorado University. All participants provided informed consent in advance of their participation in the study.

### 2.1. Experimental Design

Participants were randomized to a non-exercise control group (who were instructed to continue their usual lifestyle habits) or treatment group at a 1:1 ratio using a computerized stratified minimization sequence (Figure 1). Participants randomized to the treatment group completed a 12 week personalized exercise training program based on the American Council on Exercise (ACE) Integrated Fitness Training (IFT) model guidelines [13]. Participants within both groups completed baseline and post-program testing. Assessments of anthropometric measures, cardiometabolic risk factors, and maximal oxygen uptake (VO_2_max) were obtained at baseline and 12 week. At baseline, the talk test was performed to identify ventilatory thresholds (VT1 and VT2) for cardiorespiratory training. The procedures for all our assessments were consistent with our previous research and detailed elsewhere [13,14].

### 2.2. Personalized Exercise Training Program

The personalized exercise training program was comparable to that we have used previously [14]. Each participant consulted with a team of health and fitness professionals and was assigned a Western Colorado University undergraduate or graduate student who served as their personal trainer. The student personal trainers worked directly under the supervision of qualified MSc- and PhD-trained exercise physiologists. The exercise team designed and progressed an appropriate and safe personalized exercise program using the evidence-based ACE IFT model guidelines for both cardiorespiratory and muscular training [9]. The student personal trainers coached members during their exercise sessions, provided motivational support, engaged in spotting, and corrected exercise technique.

### 2.3. Statistical Analysis

All analyses were performed using SPSS Version 25.0 (IBM Corporation, New York, NY, USA) and GraphPad Prism 7.0. (San Diego, CA, USA). Measures of centrality and spread are presented as mean ± standard deviation (SD). Primary outcome measures included mean change in cardiorespiratory fitness and MetS z-score. Within-group comparisons were made using paired t-tests. Between-group 12 week changes were analyzed using general linear model (GLM) ANOVA for all primary outcome measures. The probability of making a type I error was set at *p* < 0.05. The coefficient of variation (CV) for MetS z-score and VO_2_max was calculated with the duplicate cardiometabolic baseline and VO_2_max measures to quantify the site-specific technical error (biological variability + measurement error) for MetS z-score and VO_2_max. To determine individual MetS z-score/VO_2_max training responsiveness delta values (Δ) were calculated (post-program minus baseline value) to establish the change (Δ) in MetS z-score/VO_2_max. The Δ in MetS z-score/VO_2_max was compared to the calculated site-specific technical error (i.e., MetS z-score/VO_2_max CV). Subsequently, participants were categorized as a responder if their Δ was greater than the site-specific technical error criterion or a non-responder if the Δ failed to exceed the site-specific criterion. The calculated site-specific technical error for MetS z-score and equated to Δ > −0.48 and %Δ > 4.9%, respectively. Chi-square (*χ*^2^) tests were used to analyze the proportion of responders and non-responders for MetS z-score and VO_2_max following the study period with a subsequent Cramer’s V test to quantify effect size. The probability of making a Type I error was set at *p* < 0.05 for all statistical analyses.

## 3. Results

The physical and physiological characteristics at baseline and 12 weeks for participants who completed the study are presented in Table 1. There were no adverse events during the intervention. After 12 weeks, the treatment group showed a significant favorable change in MetS z-score, whereas the control group demonstrated increased severity of the syndrome (between-group difference, *p* < 0.05). Following the 12-week program, cardiometabolic health worsened (*p* < 0.05) in the control group with the exception of several variables (waist circumference, total and low density lipoprotein (LDL) cholesterol, triglycerides, and blood glucose) which were unchanged (*p* > 0.05) as indicated in Table 1. In contrast, other than total and LDL cholesterol, the between-group baseline to 12 week changes in all other cardiometabolic health parameters were significantly (*p* < 0.05) more favorable in the treatment group relative to the control group (Table 1). 

The inter-individual variability in MetS z-score and VO_2_max changes following the 12-week exercise intervention (A) or control (B) period are shown in Figure 2 and Figure 3, respectively. As presented in Figure 2, the proportion of MetS z-score responders (Δ > −0.48) was significantly greater following the exercise intervention (71%, 50/70) compared to control (10%, 7/72) (between-group difference, *p* < 0.001; Cramer’s V = 0.629). The inter-individual variability in VO_2_max change also showed a similar trend. Figure 3 shows a significantly greater proportion of VO_2_max responders (Δ > 4.9%) following the exercise intervention (94%,66/70) compared to control (10%, 7/72) (between group difference, *p* < 0.001; Cramer’s V = 0.846). 

When evaluating individuals considered to have MetS according to the IDF criteria within the treatment group at baseline, there were 18 who met the criteria and 52 participants who did not. Of these 18 participants with MetS at baseline, 89% (16/18) were considered responders to a favorable MetS z-score change, whereas only 65% (34/52) in those who did not have MetS at baseline, but with no significant between-group difference (*p* = 0.06).

## 4. Discussion

This is the first study to investigate the impact of a community-based personalized training program based upon the ACE IFT model guidelines [13], combining cardiorespiratory training and muscular training, on MetS severity, and consequently T2DM and CVD risk [15]. The present study showed that this exercise training program approach can elicit significantly greater improvements in MetS severity and CRF, combined with diminished inter-individual variation in training responses, depicted by a greater number of individuals meeting the criteria for favorable changes in MetS z-score (Δ > −0.48) and VO_2_ max (Δ > 4.9%), when compared to a non-exercise control group. These novel findings are encouraging and provide insightful data for the translation of personalized exercise programs that will optimize training responsiveness at the community level.

Our study provides evidence to support the implementation of community-wide personalized exercise as a primary prevention initiative to significantly reduce the adverse effect of physical inactivity and associated comorbidities. This is clinically significant given that modest reductions in cardiometabolic risk factors induced by community initiatives have been reported to translate into the prevention of T2DM and CVD at the population level [16,17,18]. Consistent with the present findings, it has been reported that a community exercise program serves as a prophylactic against CVD via the reduction in MetS prevalence and elimination of individual MetS components [10]. This is supported by a one-year randomized study which reported a reduction in 10-year coronary heart disease risk by ~26% in African American families following a community intervention targeting multiple CVD risk factors (i.e., risk factor reduction mediated by CVD counselling and physical activity opportunities on a community level) relative to only ~3% with usual care [19]. Interestingly, the authors reported that the significant CVD risk reduction may be attributed to the favorable change in MetS components. There is also robust scientific literature demonstrating an independent, dose-response relationship between improved CRF and cardiometabolic health and reduced risk of mortality from CVD and all-causes [20,21]. Therefore, the improvement in MetS severity and CRF induced by the community-based personalized training program in the present study can be deemed clinically significant.

There is emerging evidence that suggests considerable individual variability (i.e., responders and non-responders) in exercise-induced changes in common cardiometabolic risk factors, with some individuals actually experiencing an adverse response when exposed to regular exercise. With exercise being recognized as an effective form of prevention and treatment for many chronic diseases, understanding factors associated with the variability in training responsiveness is of growing importance for health and fitness professionals. Given that one of the primary strategic mission of health organizations is to get people moving, it is paramount that health and fitness professionals have evidence-based programming options available to implement on the individual and community levels. The present findings suggest that the ACE-IFT approach is effective in improving cardiometabolic health, with almost all individuals showing a positive change in MetS severity (71%, 50/70) and CRF (94%, 66/70) beyond biological variability following a 12-week community-based program. In contrast, for those in the non-treatment group, 90% did not meet the threshold considered to be a favorable MetS z-score or CRF change. It should be noted however, that the present findings are inconsistent with previous studies using a similar exercise program which showed 100% of included participants improving CRF [8,14]. These contrasting results may simply be due to the difference in exercise program duration between the investigations (12 weeks vs. 13 weeks). Indeed, a higher volume of high-intensity exercise over the course of an intervention has been reported to induce more responders to improvements in VO_2_max relative to a lower volume exercise at high-intensity [22]. 

Consistent with our findings, Weatherwax et al. [8] also reported comparable proportion (<100%) CRF responders following 12 weeks of a similar exercise training approach. It is plausible that the less proportion of responders (<100%) to CRF and MetS z-score improvement in the present study and that of Weatherwax et al. [8] compared to other studies [8,14] may be attributed to the variability in baseline characteristics, notably the difference in baseline CVD risk of participants between studies. This is supported by our sub-analysis which showed that individuals diagnosed with MetS at baseline had a better response to MetS z-score (89%, 16/18) improvement relative to those who were not diagnosed with MetS (MetS z-score, 65% [34/52], although not statically significant. This can possibly be explained by the higher degree of sensitivity to any exercise dose in individuals with greater insulin resistance [23], a factor purported as a central mediating factor of MetS [24], relative to those without MetS. Nevertheless, the proportion of MetS z-score (>−0.48) and CRF (>4.9%) responders reported in the present study is impressive and can be deemed clinically significant given that as little as a 0.15 reduction in MetS z-score correspond to an approximate improvement in one MetS component [25]. Moreover, DeBoer et al. [4] demonstrated that a similar reduction in MetS z-score of 0.62 after a one-year lifestyle intervention is associated with reduced incident of T2DM and CVD risk. Finally, the similar pattern of changes evident between MetS severity and CRF is also not surprising given that CRF has repeatedly been reported as an antidote against individual risk factors associated with MetS [26].

### Limitations

While the undergraduate and graduate student personal trainers were under direct supervision there may still have been the element of error in administering the tests and exercise training program. No dietary information was taken from the participants during the 12 week program which may have provided further, tangible information to help explain some of the results. Lastly, there were no long-term analysis done so long-term benefits are unknown. 

## 5. Conclusions

Our current findings provide critical translational evidence demonstrating personalized exercise programming based upon the ACE IFT model guidelines can be successfully implemented within the community setting to reduce the severity of MetS and consequently T2DM and CVD risk [15]. Notably, this community-based intervention appears to be more effective in those already diagnosed with MetS relative to those without MetS. This is an important finding given that individuals with MetS are known to be at higher risk of T2DM and CVD compared to those without the condition.

## Figures and Tables

**Figure 1 ijerph-16-04855-f001:**
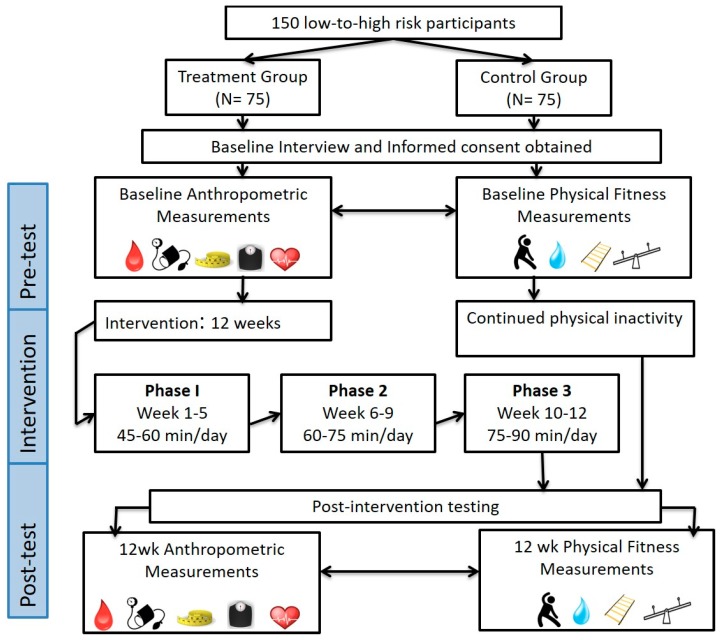
Study protocol for community based individualized exercise prescription. Baseline testing legend: Lipid testing 
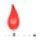
; Submaximal VO_2_max test (talk test) 
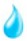
; Balance test 
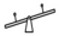
; 8 feet up and go 
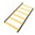
; Shoulder rotation, sit and reach, trunk rotation 
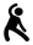
; Blood pressure 
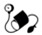
; Resting heart rate 
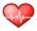
; BMI 
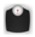
; Waist circumference 
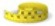
.

**Figure 2 ijerph-16-04855-f002:**
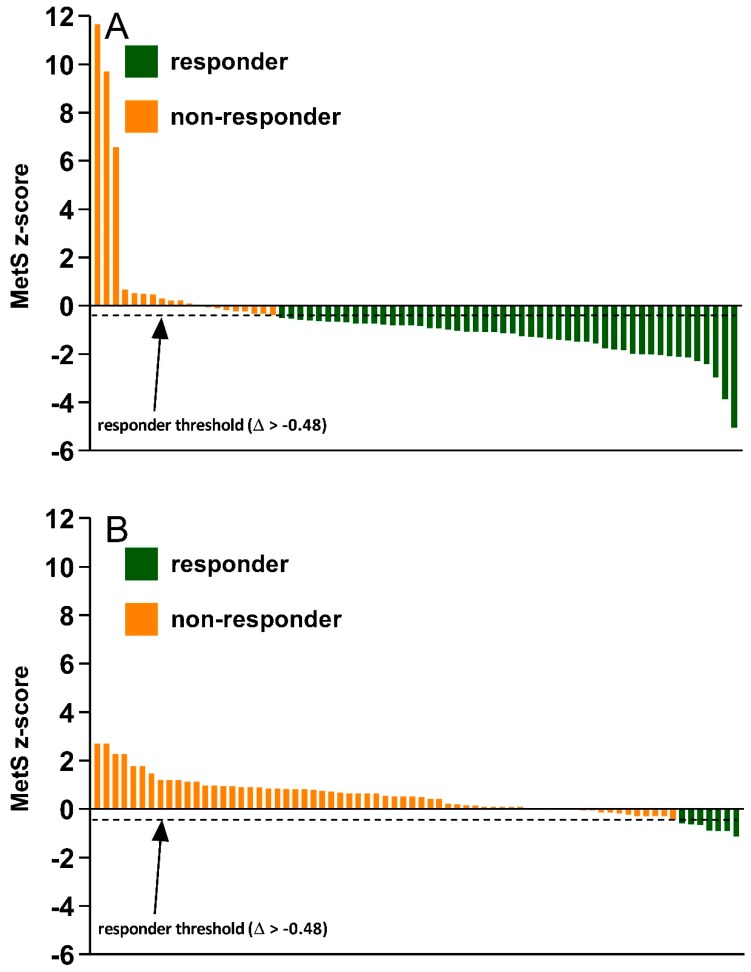
Inter-individual variability in MetS z-score change following the exercise intervention (**A**) or control period (**B**).

**Figure 3 ijerph-16-04855-f003:**
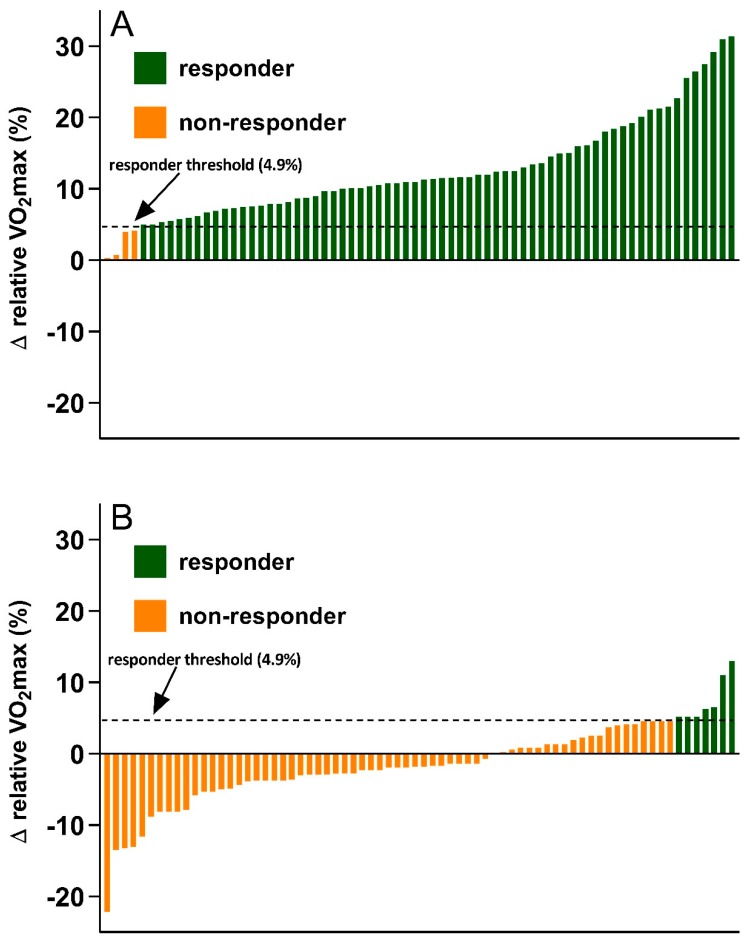
Inter-individual variability in VO_2_max change following the exercise intervention (**A**) and control (**B**) period.

**Table 1 ijerph-16-04855-t001:** Physical and physiological characteristics at baseline and 12 weeks for control and treatment groups (values are mean ± SD).

Outcome Variable	Control Group (*N* = 72)		Treatment Group (*N* = 70)	
	Baseline	Post-Program	Baseline	Post-Program
Age (yr)	45.6 ± 12.5	-------	46.6 ± 16.7	-------
Body mass (kg)	75.5 ± 12.3	75.7 ± 12.0 *	77.3 ± 18.7	76.7 ± 18.4 *^, †^
Waist circumference (cm)	82.4 ± 8.8	82.7 ± 8.6	84.0 ± 14.2	83.1 ± 12.9 *^, †^
Systolic BP (mm Hg)	119.0 ± 11.0	121.2 ± 9.6 *	122.6 ± 14.1	117.4 ± 13.1 *^, †^
Diastolic BP (mm Hg)	79.4 ± 8.4	81.4 ± 6.6 *	79.7 ± 9.7	77.3 ± 7.7 *^, †^
Total cholesterol (mg·dL^−1^)	201.3 ± 40.0	204.4 ± 37.5	187.5 ± 39.1	185.1 ± 37.7
HDL cholesterol (mg·dL^−1^)	50.7 ± 18.2	49.4 ± 16.5 *	54.2 ± 17.9	57.8 ± 15.9 *^, †^
LDL cholesterol (mg·dL^−1^)	119.9 ± 37.7	122.0 ± 36.3	107.2 ± 32.9	100.6 ± 31.1
Triglycerides (mg·dL^−1^)	130.0 ± 64.3	136.1 ± 67.2	110.8 ± 54.4	104.5 ± 45.7 ^†^
Blood glucose (mg·dL^−1^)	93.1 ± 9.0	94.8 ± 9.1	92.5 ± 8.6	89.7 ± 7.0 *^, †^
VO_2_max (mL·kg^−1^·min^−1^)	29.0 ± 6.1	28.4 ± 5.8 *	31.4 ± 7.9	35.0 ± 8.0 *^, †^
MetS z-score	−4.15 ± 4.01	−3.68 ± 4.07 *	−3.52 ± 3.82	−4.12 ± 3.24 *^, †^

BP, blood pressure; HDL, high density lipoprotein; LDL, low density lipoprotein; MetS, metabolic syndrome; VO_2_max, maximal oxygen uptake. * Within-group change is significantly different from baseline, *p* < 0.05; ^†^ Change from baseline is significantly different from control group, *p* < 0.05.

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
