# Peer review of "Inter-Individual Variability in Metabolic Syndrome Severity Score and VO2max Changes Following Personalized, Community-Based Exercise Programming"

_ijerph, 2019, doi:10.3390/ijerph16234855_

Round 1

Reviewer 1 Report

Seward and co-workers examine the effectiveness of a personalized, community-based exercise program at reducing MetS severity. This is an innovative study, however methodological and statistical problems militate against the publication of the present manuscript:

Instead of a number of separate t-tests on the data, 2 (groups: Exercise/control) x 2 (Tests: Baseline/Post-program) ANOVAs should be performed. Moreover, not only significance levels but also effect sizes should be reported to illustrate the practical value of possible improvements. Please provide results for the power analysis that indicate your sample size was appropriate. In your case, if you want to detect a large effect size in your primary outcomes (mean change in cardiorespiratory fitness and metabolic syndrome z-score), considering the following design specifications: α= 0.05; (1-β) = 0.8; effect size f = 0.5; test family = F test and statistical test= 2 way ANOVA for repeated measures, the sample size should be of 38 subjects, 19 in each group. The authors claim to have included men and women in the study without specifying the number. If this is unbalanced, the sex factor should be considered in the ANOVA.

Additional comments:

For any equipment used to collect data that contributed to an analyzed variable, provide the model, manufacturer, and country, unless it is contained in previously published methodology you have cited. Authors should provide sufficient information that the reader can assess the method used to generate the random allocation sequence and the likelihood of bias in group assignment I understand that your intervention protocol was used in a previous study [14], but key information about the program should be reported.

Overall, the study may have the potential to add meaningful information to the current body of literature.

Author Response

Reviewer #1

Instead of a number of separate t-tests on the data, 2 (groups: Exercise/control) x 2 (Tests: Baseline/Post-program) ANOVAs should be performed. Moreover, not only significance levels but also effect sizes should be reported to illustrate the practical value of possible improvements. Please provide results for the power analysis that indicate your sample size was appropriate. In your case, if you want to detect a large effect size in your primary outcomes (mean change in cardiorespiratory fitness and metabolic syndrome z-score), considering the following design specifications: α= 0.05; (1-β) = 0.8; effect size f = 0.5; test family = F test and statistical test= 2 way ANOVA for repeated measures, the sample size should be of 38 subjects, 19 in each group. The authors claim to have included men and women in the study without specifying the number. If this is unbalanced, the sex factor should be considered in the ANOVA.

RESPONSE: ANOVA’s were performed in our original analyses and we have revised the paper accordingly – thank you. We have added effect sizes (via Cramer’s V) to our revised paper – thank you. We did not perform a power analysis a priori. Considering all our main analyses were significant a post hoc power analysis is deemed unnecessary. It has been previously reported (Bouchard & Rankinen, 2001) that age, sex, race, and initial training values do not influence the heterogeneity in the individual response to exercise training, therefore we did not include sex as a covariate. Thank you for all of your comments and excellent suggestions.   

For any equipment used to collect data that contributed to an analyzed variable, provide the model, manufacturer, and country, unless it is contained in previously published methodology you have cited. Authors should provide sufficient information that the reader can assess the method used to generate the random allocation sequence and the likelihood of bias in group assignment I understand that your intervention protocol was used in a previous study [14], but key information about the program should be reported.

RESPONSE: Details of our intervention have been published previously and we indicate that in our paper. Additionally, key information about the intervention are presented in Figure 1. Participants were randomly allocated to either the control or treatment group at a 1:1 ratio using a computerized stratified minimization sequence and this has been added per your comment to the revised paper. Thank you again for all your comments and excellent suggestions.

Reviewer 2 Report

Abstract:  number of words should be according to guidelines of the journal Please explain the abbreviations within the tables under each table. Some recent suggested references which could be used which have shown important of meeting physical activity levels especially in the same target populations: The Associations between Mental Well-Being and Adherence to Physical Activity Guidelines in Patients with Cardiovascular Disease: Results from the Scottish Health Survey Factors Associated with Meeting Current Recommendation for Physical Activity in Scottish Adults with Diabetes

Author Response

Reviewer #2

Abstract:  number of words should be according to guidelines of the journal Please explain the abbreviations within the tables under each table. Some recent suggested references which could be used which have shown important of meeting physical activity levels especially in the same target populations: The Associations between Mental Well-Being and Adherence to Physical Activity Guidelines in Patients with Cardiovascular Disease: Results from the Scottish Health Survey Factors Associated with Meeting Current Recommendation for Physical Activity in Scottish Adults with Diabetes

RESPONSE: The abstract has been revised to be under the 200-word limit. An explanation of abbreviations has been added as suggested. Thank you for the suggested references; however, we have elected not to integrate as these studies focused on satisfying physical activity recommendations whilst our paper examined the responsiveness to personalized exercise training. Thank you for your excellent suggestions.

Reviewer 3 Report

In this study, authors evaluated the inter-individual variability of changes in VO2max and MetS z-score after training program based on ACE-IFT model guidelines. According to their data, program was effective to improve VO2max and MetS z-score and there was large inter-individual variability in its effectiveness. This program seemed more effective in those already diagnosed with MetS.

P2, lines 72-77: Were there exclusion criteria? How did authors treat subjects with medications for oral hypoglycemic agents, anti-hypertensive agents, and anti-hyperlipidemic agents? P4, lines 130-131 & Table 1: In Figure 1, participants of both groups are 75. But, Table 1 showed n=72 in control group and n=70 in treatment group. Please give me the reason of drop-out. Furthermore, If authors can, please make intention-to-treat analysis as well as per-protocol analysis, including drop out participants.

Author Response

Reviewer #3

P2, lines 72-77: Were there exclusion criteria? How did authors treat subjects with medications for oral hypoglycemic agents, anti-hypertensive agents, and anti-hyperlipidemic agents? P4, lines 130-131 & Table 1: In Figure 1, participants of both groups are 75. But, Table 1 showed n=72 in control group and n=70 in treatment group. Please give me the reason of drop-out. Furthermore, If authors can, please make intention-to-treat analysis as well as per-protocol analysis, including drop out participants.

RESPONSE: We have listed our inclusionary criteria, which included nonsmoking, low-to-high risk individuals, and physically inactivity. Otherwise, there were no specific exclusionary criteria. Medications were unaccounted for in the present study. There were no drop outs once participants commenced with training. However, five participants in the treatment group and three participants in the control group did not return after baseline testing for reasons unknown. Because all participants who started the intervention completed the intervention, we did not complete an intention-to-treat analysis. Thank you for your excellent suggestions.